# Orientation Controllable RCS Enhancement Electromagnetic Surface to Improve the Road Barriers Detectability for Autonomous Driving Radar

**DOI:** 10.3390/s25134048

**Published:** 2025-06-29

**Authors:** Yanbin Chen, Tong Wang, Qi Liu, Haochen Wang, Cheng Jin

**Affiliations:** 1School of Cyberspace and Technology, Beijing Institute of Technology, Beijing 100086, China; wt15536840224@163.com (T.W.); liuqi@bit.edu.cn (Q.L.); wanghaochen@bit.edu.cn (H.W.); jincheng@bit.edu.cn (C.J.); 2Space Star Technology Co., Ltd., Beijing 100086, China

**Keywords:** Van Atta, radar cross section enhancement, reconfigurable array, retrodirective array

## Abstract

An orientation controllable radar cross section (RCS) enhancement surface is presented in this paper, which can be used to improve the road pile detectability of on-board microwave radar for autonomous driving system. In addition, the RCS enhancement orientation can be controlled in a specified direction without interfering with other microwave systems. We first designed a modified one-dimensional VanAtta array with adjustable phase for retrodirective backtracking the incoming electromagnetic waves, which can achieve wide-angle RCS enhancement. Then, we arranged the one-dimensional VanAtta array in another dimension forming a two-dimensional array, enabling adjustable orientation RCS enhancement due to the controllable phase of the reflected electromagnetic waves. We designed, manufactured, and tested a 4 × 8 array to validate the theory and assess the design’s feasibility. Finally, six orientation controllable VanAtta arrays were mounted on the outside surface of a cylinder road barrier, and measurements demonstrated that RCS enhancement of over 10 dB have been achieved compared to the same pile with perfect electric conductor surface.

## 1. Introduction

Automatic driving is an important direction in the development of the current automobile industry, which has made remarkable progress in recent years, but the auto drive system still faces difficulties, and many test vehicle accidents are caused by the system’s failure to accurately identify obstacles [1]. In order to improve the detection ability of microwave radar for low RCS targets (such as plastic billboards and pedestrians), we propose an electromagnetic reflection enhancement surface with electromagnetic wave backtracking function. The key technology of surface design is electromagnetic wave backtracking electromagnetic surface technology, which is similar to RCS enhanced electromagnetic surface technology in radar systems, both of which are enhanced reflections of radar waves received by electromagnetic surfaces. RCS-enhanced surface is an electromagnetic surface that reflects electromagnetic waves in the direction of the incoming wave. This surface has application value in fields such as the automatic driving of cars [2,3,4], target tracking [5,6], wireless communication [7,8], and wireless energy transmission [9].

Plastic barriers are typically used to enclose the area around a road under construction, and affixing our designed surface to these barriers can enhance the detection capability of vehicle-mounted microwave radar for barriers, thereby promoting the safety of the construction site. The scene depicted in Figure 1 shows a barrier located above a slope. On the left side of the barrier is a flat road, and the RCS enhancement direction on the left side of the barrier should be horizontal. On the right side of the barrier is a downhill road segment; therefore, the RCS enhancement direction on the right side of the barrier should be diagonal and downward. Additionally, the direction of RCS enhancement on the electromagnetic surface is parallel to the road surface, thus avoiding interference with other microwave systems. To meet the requirements of the mentioned application scenario, we need a surface that provided wide-angle RCS enhancement horizontally while allowing for vertical adjustment of the RCS enhancement direction, which was then conformal on a cylindrical obstacle and adjusted to match the application based on the road environment in which the obstacle was located.

A special type of radar reflector has been proposed [2] to improve the detectability of low RCS targets such as pedestrians and plastic billboards in the field of autonomous driving, the one-dimensional Van Atta array that enhances RCS only in the horizontal direction. However, it has limitations when the height of the billboard varies or when it is applied to billboards located on the sides of roads with slopes, such as at corners of underground garages. On the other hand, a two-dimensional Van Atta array [10] should be suitable to provide RCS enhancement in both the horizontal and vertical direction. However, its omni-directional enhancement characteristic makes it susceptible to interfering with other microwave systems. Then, we propose an electromagnetic backtracking surface combining Van Atta and phase gradient surface, which has the following advantages: (1) The horizontal RCS enhancement angle range is very wide, and it only reflects high gain echoes in the direction of the incoming wave, without interference to other vehicles in the horizontal direction. (2) The vertical direction can adjust the angle to adapt to various road conditions, and the adjustment will not interfere with wireless devices at other heights. (3) The thickness of the electromagnetic surface is relatively thin, making it easy to form cylindrical or conical shapes.

The article is organized as follows. Section 2 discusses the principles of RCS Enhancement. In Section 3, the proposed surface is fabricated and measured. Finally, Section 4 concludes this article.

## 2. Principles of RCS Enhancement

In order to meet the aforementioned application requirements, we proposed a one-dimensional reflect array for the first step. This array is capable of providing wide-angle monostatic RCS enhancement in the horizontal direction. Additionally, the one-dimensional array features adjustable phase of reflected electromagnetic waves. Only by making the phase of reflected electromagnetic waves adjustable can the array possess the characteristic of adjustable RCS enhancement direction. In the second step, the proposed one-dimensional array is arranged vertically to form a two-dimensional array. By controlling the phase of the reflected electromagnetic waves within the array, adjustable RCS enhancement direction can be achieved in the vertical direction. The principle of the one-dimensional array will be discussed in the following.

### 2.1. One-Dimensional Retro-Reflective Array

The proposed one-dimensional electromagnetic retro-reflective array, depicted in Figure 2, operates as follows: The electromagnetic waves received by the radiating unit (0) are transmitted from the antenna unit (I) after passing through both the feedline and phase shifter. Similarly, the electromagnetic waves received by the radiating unit (I) are transmitted from the antenna unit (0) after passing through the feedline and phase shifter. The same principle applies to the corresponding units (1) and (I − 1). When the feedline lengths between antenna pairs are equal, and the phase shifter induces an identical phase variation, the one-dimensional array demonstrates electromagnetic retroreflective characteristics owing to its phase conjugation property. As shown in Figure 2, the red dashed line represents the incident electromagnetic wavefront, while the green dashed line represents the reflected electromagnetic wavefront. By adjusting the phase of all phase shifters in the one-dimensional array, each radiating element emits electromagnetic waves with a phase that is advanced or delayed to the same extent. This effectively alters the phase of the emitted electromagnetic wavefront, which is visually represented by the vertical displacement of the green dashed line in the figure.

Referring to the one-dimensional Van Atta array’s mono-static RCS scattering field equation [11], we obtain the monostatic RCS scattering field equation of the above one-dimensional array as shown in (1). According to the equation, it can be inferred that the monostatic RCS of the array at different angles is only related to the array’s unit radiation pattern.(1)$$E_{m}\left(\theta \right)=\frac{e^{-jK_{0^{r}}}}{r}(C_{f}E_{0}e^{-j\left(I-1\right)K_{0}dcos\left(\theta \right)}e^{-jK_{tl}L_{tl}}e^{-j2\psi Phaser})IG\left(\theta \right)$$
where $K_{0}$ denotes the wave number, $C_{f}$ represents the feed factor, $E_{0}$ represents the amplitude of the incident electric field, I is the total number of elements, d represents the spacing between elements, θ is the direction of the incident wave, $K_{tl}$ represents the wave number in the transmission line, $L_{tl}$ represents the length of the transmission line, ψPhaser is the phiPhaser, and G(θ) is the gain of the element in the θ direction. The normalized monostatic RCS of the one-dimensional array is shown in Figure 3, where the power pattern of the antenna element is represented by $U=U_{m}cos\theta ,\;(0\;\leq \;\theta \;\leq \;\pi /2,\;0\;\leq \;\varphi \;\leq \;2\pi )$, and $U_{m}$ is the maximum radiation intensity.

It is important to emphasize that the phase of all phase shifters in the 1D array is the same, and all phase shifters in the entire 1D array only need to occupy one control unit, which greatly saves cost and design complexity.

### 2.2. RCS Enhancement Principle

The aforementioned one-dimensional array is arranged in another dimension to form a two-dimensional array. The following section derives the principle behind RCS enhancement achieved by the two-dimensional array. Assuming the antenna units are arranged in an equally spaced configuration, with a total of I rows and J columns, as depicted in Figure 4, the phase distribution of the received signal [11] is(2)$$\Phi_{\left(m,n\right)}^{r}=K_{0}(md_{x}sin\theta_{in}cos\varphi_{in}+nd_{y}sin\theta_{in}sin\varphi_{in})$$
where $K_{0}$ denotes the wave number, $d_{x}$ represents the array spacing of the antenna unit along the X-axis, $d_{y}$ is the array spacing along the Y-axis, and the incoming wave direction is given by $\left(\theta_{in},\;\varphi_{in}\right)$. The indices (m, n) indicate the m-th row and the n-th column. Assuming(3)$$\left\{\begin{array}{c} \partial_{x}=K_{0}d_{x}sin\theta_{in}cos\varphi_{in}\\ \partial_{y}=K_{0}d_{x}sin\theta_{in}sin\varphi_{in} \end{array}\right.\;$$

$\partial_{x}$ represents the phase gradient of the unit along the X-axis, $\partial_{y}$ represents the phase gradient of the unit along the Y-axis. Then [11](4)$$\Phi_{\left(m,n\right)}^{r}=m\partial_{x}+{n\partial }_{y}$$
subsequently, we consider any individual unit denoted by (m, n)(0 ≤ m ≤ I, 0 ≤ n ≤ J). The electromagnetic wave received by the (m, n) unit undergoes transmission through a dedicated line and two-phase shifters, ultimately being emitted by the (I−m, n) units. The resulting emission phase is(5)$$\Phi_{\left(I-m,n\right)}^{t}=\Phi_{\left(m,n\right)}^{r}-K_{tl}L_{tl}-2\psi Phaser(n)$$
here, $K_{tl}$ represents the wave number of the electromagnetic waves in the microstrip line, while $L_{tl}$ represents the length of the microstrip line.(6)$$\psi Phaser\left(n,\theta_{p}\right)=\psi_{0}+K_{0}nd_{y}sin\theta_{p}$$

$\psi_{0}$ is the initial phase of the phase shifter, $K_{0}nd_{y}$ sin $\theta_{p}$ represent the phase of the phase shifter, where $\theta_{p}$ is an assumed angle. The reason the phase of the phase shifter is written as $K_{0}nd_{y}$ sin $\theta_{p}$ is to facilitate the derivation of subsequent formulas. (Specifically, when electromagnetic waves are incident on the electromagnetic surface along the CO direction as shown in Figure 5, $\theta_{p}$ is numerically equal to $\theta_{rot}$. At this point, the phase shifter precisely compensates for the phase gradient along the Y-axis, and the electromagnetic surface reflects the electromagnetic wave along the OC direction.)

On the contrary, when using (I−m, n) as the receiving unit, the electromagnetic wave is radiated by the (m, n) unit, and the phase distribution fed into the transmitting array is(7)$$\Phi_{\left(m,n\right)}^{t}=\Phi_{\left(I-m,n\right)}^{r}-K_{tl}L_{tl}-2\psi Phaser(n)$$

When adding the phase distribution of the incident signal to the phase distribution of the transmitted signal and simplify, we have the following:(8)$$\Phi_{\left(m,n\right)}^{r}+\Phi_{\left(m,n\right)}^{t}={I\partial }_{x}+2n\partial_{y}-2nK_{0}d_{y}sin\theta_{p}-K_{tl}L_{tl}-2\psi_{0}$$
where $K_{tl}$, $L_{tl}$, and $\psi_{0}$ are constants, assuming $C=-K_{tl}L_{tl}-2\psi_{0}$. We rewrite the above equation as(9)$$\Phi_{\left(m,n\right)}^{r}+\Phi_{\left(m,n\right)}^{t}={I\partial }_{x}+2nK_{0}d_{y}(sin\theta_{in}sin\varphi_{in}-sin\theta_{p})+C$$

When $sin\theta_{p}$ = $sin\theta_{in}sin\varphi_{in}$, which means that the phase difference in the unit along the Y-axis is compensated by the phase shifter, then(10)$$\Phi_{\left(m,n\right)}^{r}+\Phi_{\left(m,n\right)}^{t}={I\partial }_{x}+C$$
since Equation (10) does not include the variables m and n, it means that regardless of the element taken from any row or column of the array, the sum of the phase of the electromagnetic wave emitted and received by that element is equal to the sum of the phase of the electromagnetic wave emitted and received by any other element. Therefore, the electromagnetic wave can be retroacted, achieving the effect of RCS enhancement.

### 2.3. Coordinate Transformation

The relationship between the phase shift angle of the phase shifter and the direction of the incoming wave is(11)$$\psi Phaser\left(n,\theta_{in},\varphi_{in}\right)=\psi_{0}+K_{0}nd_{y}sin\theta_{in}sin\varphi_{in}$$

When the incident electromagnetic wave is incident within the $Y=tan\;\theta_{rot}Z$ plane, assuming the angle between the incident electromagnetic wave and the Y−O−Z plane is α, the direction of the incident electromagnetic wave can be determined by $\theta_{rot}$ and α. As the electromagnetic wave is in the plane $Y=tan\;\theta_{rot}Z$, for a given incident wave angle $\theta_{in}^{\prime }(\theta_{rot}^{{}^{\circ}}\;\leq \;\theta_{in}^{\prime }\;\leq 90^{{}^{\circ}}$), the incident wave angle $\varphi_{in}^{\prime }$ can be determined:(12)$$\varphi_{in}^{\prime }\left(\theta_{rot},\theta_{in}^{\prime }\right)=arctan(\frac{\pm tan\theta_{rot}cos\theta_{in}^{\prime }}{\sqrt{1-(1+2{tan}^{2}\theta_{rot}){cos}^{2}\theta_{in}^{\prime }}})$$(13)$$\alpha \left(\theta_{rot},\theta_{in}^{\prime }\right)=arcsin(\pm \sqrt{1-(1+{tan}^{2}\theta_{rot}){cos}^{2}\theta_{in}^{\prime }})$$

Finally, the relationship between ($\theta_{in},\varphi_{in}$) and ($\theta_{rot}$, α) is given by the following equations:(14)$$\left\{\begin{array}{c} \varphi_{in}(\theta_{rot},\;\alpha )=arctan(\frac{\pm tan\theta_{rot}}{tan\alpha \sqrt{1+{tan}^{2}\theta_{rot}}})\;\\ \theta_{in}(\theta_{rot},\;\alpha )=arccos(\frac{\pm cos\alpha }{\sqrt{1+{tan}^{2}\theta_{rot}}}) \end{array}\right.\;$$

When electromagnetic waves are incident in the $Y=tan\theta_{rot}Z$ plane and the phase of the phase shifters in the array remains fixed, the sum of the phases of the electromagnetic waves received and transmitted by the array elements is no longer constant when the angle α changes. There exists a difference in the sum of the phases of electromagnetic waves received and transmitted between the elements, leading to a change in the phase conjugate characteristics of the array’s transmission and reception of electromagnetic waves. The difference in maximum phase between the sum of the phases of electromagnetic waves received and transmitted among the elements is expressed in (15), which can be used to describe the degree of phase mismatch in the array.(15)$${Phaser}_{mismatch}=2nK_{0}d_{y}(sin\theta_{in}sin\varphi_{in}-sin\theta_{p})$$

Substituting Equation (14) into Equation (15) yields:(16)$${Phaser}_{mismatch}\left(\theta_{rot},\alpha \right)\;\;=2nK_{0}d_{y}(\sin\left(\arccos\left(\frac{\pm cos\alpha }{\sqrt{1+{tan}^{2}\theta_{rot}}}\right)\right)\;sin\left(\arctan\left(\frac{\pm tan\theta_{rot}}{tan\alpha \sqrt{1+{tan}^{2}\theta_{rot}}}\right)\right)-sin\theta_{p})$$

### 2.4. Calculation and Discussion

For an incident wave in the $Y=tan\;15^{{}^{\circ}}Z$ plane with a phase shifter phase ψPhaser = $\psi_{0}$+$K_{0}nd_{y}sin15^{{}^{\circ}}$, the degree of phase mismatch increases as α increases.

If a difference of over $90^{{}^{\circ}}$ between the sum of the phases of electromagnetic waves received and transmitted among the elements is taken as the criterion for complete phase mismatch, the enhancement angle range is ±$28^{{}^{\circ}}$, as shown in Figure 6a. As shown by the red curve in Figure 6a, when the incident wave is at α = $0^{{}^{\circ}}$, by adjusting the phase of the phase shifter and adding appropriate phase compensation to the array, the maximum difference of $90^{{}^{\circ}}$ in the sum of the phases of electromagnetic waves received and transmitted among the elements occurs. As α changes from $0^{{}^{\circ}}$ to $45^{{}^{\circ}}$, this difference changes from +$90^{{}^{\circ}}$ to −$90^{{}^{\circ}}$, indicating a gradual decrease and then increase in the degree of phase mismatch. The enhanced angle range α is expanded to ±$40^{{}^{\circ}}$. If overcompensated, there is significant phase mismatch at α = $0^{{}^{\circ}}$, and the enhanced angle range α is from $30^{{}^{\circ}}$–$50^{{}^{\circ}}$. When the incident wave is in the $Y=tan\;30^{{}^{\circ}}Z$ plane, the rate of phase mismatch increases. After applying appropriate phase compensation, the enhanced angle range for RCS is approximately ±$29^{{}^{\circ}}$, as shown in Figure 6b.

## 3. Design of the Applications

According to the theoretical part, we need to design an antenna unit with a phase shifter, which should meet the phase shifting range of 0–$180^{{}^{\circ}}$ and the phase should be continuously adjustable, so that the electromagnetic wave received by such an antenna unit is transmitted through two phase shifters through another antenna unit. In this way, the phase change range of the emitted electromagnetic wave reaches 0–$360^{{}^{\circ}}$.

### 3.1. Unit Design

The design of the microstrip antenna unit is shown in Figure 7. The top layer of the antenna unit consists of a radiating patch. Below that is a 1.524 mm-thick dielectric layer made of RO4350B material, with a relative permittivity of 3.66. Underneath the dielectric is the antenna’s ground plane, which has a slot for coupling and feeding the antenna. Below the ground plane is another dielectric layer, 0.508 mm thick, also made of RO4350B material. At the bottom is the microstrip line layer, which mainly includes a phase shifter and a feeding microstrip line. The detailed dimensions for each component can be found in Table 1.

The phase shifter used in the design is an analog phase shifter, which consists of a microstrip coupled line and three variable capacitor diodes [12]. In the design of a phase shifter, we should pay attention to the design of the coupling line. The odd mode impedance of the coupling line in this design is 45.6 Ω, and the even mode impedance is 80.1 Ω. The length of the coupling line shall be close to one eighth of the wavelength of the operating frequency. After optimization design, the length of the coupling line determined in this design is 4.7 mm. The model number of the variable capacitor diode is MA46H120. When the DC bias voltage varies from 0 V to 14 V, the capacitance of the variable capacitor diode changes accordingly from 1.15 pF to 0.15 pF. During operation, the variable capacitor diodes are biased in the reverse direction. Therefore, the anode of the variable capacitor diodes is connected to the ground through a via hole, while the cathode is connected to the microstrip coupled line, as shown in Figure 7b.

In the PCB manufacturing process, it can be challenging to drill blind vias. To facilitate machining and reduce costs, a through-hole for grounding is drilled through the entire board. To avoid connecting the radiating patch to the ground plane, a circular area with a radius of 0.7 mm is removed from the copper on the radiating patch.

During the modeling and simulation process in Ansys HFSS, the variable capacitor diode is set to be connected in series with a 2 Ω resistor and a capacitor. The simulated S11 parameter of the unit is shown in Figure 8. It can be observed that when the capacitance of the variable capacitor diode varies between 0.15 pF to 1.15 pF, the S11 parameter of the unit undergoes some changes. However, regardless of the operating state of the variable capacitor diode, the S11 parameter of the unit remains below −10 dB within the frequency range of 9–10 GHz. The curve indicates that the unit has two resonance peaks, one caused by the radiating patch and the other caused by the phase shifter. The combined effect of these two peaks keeps the S11 parameter of the unit below −10 dB within the 9–10 GHz frequency range.

The phase shift range of the unit is shown in Figure 8b, taking 9.5 GHz as an example, when the capacitance of the variable capacitor diode changes from 0.15 pF to 1.15 pF, the phase can continuously vary from −$145.1^{{}^{\circ}}$ to −$324.8^{{}^{\circ}}$, resulting in a phase shift range of approximately 180°. Figure 9 shows that the realized gain of the unit is 3.74 dB, with a beam width of ±$45^{{}^{\circ}}$ at 9.5 GHz.

### 3.2. One-Dimensional Retro-Reflective Array

Connect four antenna units with equal length microstrip lines to form a 1 × 4 Van Atta array as shown in Figure 10a. Lead out the DC bias line from the edge of the microstrip line and add a sector branch on the bias line to prevent the microwave signals from entering the DC bias circuit. Connect two DC bias lines to a surface mount pad.

When DC voltage excitation is applied to the pad, the excitation voltage received by the four antenna units is the same, and the phase shifting angle of the phase shifter on each unit is the same. According to (8), when the electromagnetic wave is incident in the plane $\varphi_{in}=0$, select any unit (m, 0)(0 ≤ m ≤ I) and the sum of its receiving and transmitting phases is shown as follows:(17)$$\Phi_{\left(m,0\right)}^{r}+\Phi_{\left(m,0\right)}^{t}={I\partial }_{x}-K_{tl}L_{tl}-2\psi_{0}$$

The variables I, $\partial_{x}$, $K_{tl}$, $L_{tl}$, and the initial phase $\psi_{0}$ in (17) are all independent of m. Therefore, no matter what value m takes, the result of the equation is always the same. In other words, no matter which unit is chosen, the sum of the transmitted and received electromagnetic wave phases is equal to that of the other units. If this condition is satisfied, it proves that the 1 × 4 array has the ability to automatically backtrack electromagnetic waves. Figure 10b shows the simulated monostatic RCS in Ansys HFSS, depicting the monostatic RCS of the 1 × 4 Van Atta array when the incident wave is directed at the φ = $0^{{}^{\circ}}$ plane. The Van Atta array exhibits an approximate 5 dB enhancement in RCS within the range of $\theta_{in}$ = ±$60^{{}^{\circ}}$. However, when the electromagnetic wave is incident at ($\varphi_{in}$ = $0^{{}^{\circ}}$, $\theta_{in}$ = $0^{{}^{\circ}}$), the monostatic RCS of the array is 3.6 dB lower than that of the PEC, which is mainly due to the insertion loss of the phase shifter, the transmission loss of the microstrip line, and the antenna gain. As a result, a portion of the electromagnetic energy received by the receiving antenna is lost after passing through the phase shifter and microstrip transmission line. When the transmitting antenna transmits, it cannot concentrate all the energy in the direction of the incoming wave but rather transmits electromagnetic energy in all directions according to the radiation pattern.

### 3.3. Two-Dimensional Retro-Reflective Array

The 1 × 4 array mentioned above is arranged along the Y-axis into eight columns, forming a 4 × 8 array. Each column of the Van Atta array has a surface mount pad that is connected to two DC bias lines. By applying a DC bias voltage to this pad, the operating state of the phase shifter varactor diode can be changed, thereby altering the phase of the phase shifter and the phase of the electromagnetic waves emitted by the Van Atta array in that column. Each column of the array only requires one bias voltage for control, which means that the phase shift angle control range of each phase shifter on a single column of the array is the same. However, the phase shift angles of the phase shifters between different columns can be different and adjusted as needed in Figure 11. 

By adjusting the phase shifters of the Van Atta array to align with the direction of the incident electromagnetic wave ($\theta_{in}$, $\varphi_{in}$) as per (11), the 4 × 8 array can effectively track the electromagnetic wave along the incident wave direction, leading to enhanced RCS. The simulated monostatic RCS results in Ansys HFSS are illustrated in Figure 12.

In Figure 12a, the simulated monostatic RCS of the 4 × 8 array is shown when subjected to TE polarized electromagnetic waves in the Phi = $0^{{}^{\circ}}$ plane. The simulation indicates a significant RCS enhancement exceeding 15 dB within $Theta=\pm 60^{{}^{\circ}}$, where the phase shift angle of each phase shifter in the array remains consistent. By adjusting the phase of the emitted electromagnetic waves from each column of the 1 × 4 Van Atta array based on the principle of phase gradient, the 4 × 8 array can reflect the electromagnetic waves back towards the incident direction within the Phi = $90^{{}^{\circ}}$ plane. Figure 12b illustrates the effectiveness of this phase gradient strategy in enhancing RCS when exposed to incident electromagnetic waves at Phi = $90^{{}^{\circ}}$, Theta = $90^{{}^{\circ}}$, $15^{{}^{\circ}}$, $30^{{}^{\circ}}$, $45^{{}^{\circ}}$, and $60^{{}^{\circ}}$.

The results demonstrate an RCS enhancement effect exceeding 12 dB within the Theta range of $0^{{}^{\circ}}$–$60^{{}^{\circ}}$.

To facilitate simulation in Ansys HFSS, the array is rotated by an angle $\theta_{rot}$ along the X-axis, while the incident wave remains set to enter in the X−O−Z plane. In this configuration, the array is equivalent to be stationary, while the incident wave enters in the Y = $tan\theta_{rot}Z$ plane. The angle between the incident wave and the Y−O−Z plane is denoted as α. The relationship between the commonly used expression of incident waves ($\theta_{in}$, $\varphi_{in}$) and the proposed expression ($\theta_{rot}$, α) is shown in (14). Figure 12c depicts the enhancement of RCS for the 4 × 8 array when electromagnetic waves are incident in the $Y=tan15^{{}^{\circ}}Z$ plane. The red curve illustrates the variation in monostatic RCS of the array with respect to the α angle, assuming no phase compensation. After adjusting the phase shifters for moderate compensation, the RCS decreases due to a certain degree of phase mismatch near α = $0^{{}^{\circ}}$.

However, the enhancement range expands, as indicated by the blue curve. If overcompensated, the degree of phase mismatch near α = $0^{{}^{\circ}}$ increases, resulting in further reduction in RCS values, as shown by the green curve in the graph. The phase of each column’s phase shifter under different compensation states are shown in Table 2. When electromagnetic waves are incident within the $Y=tan\;30^{{}^{\circ}}Z$ plane, the proposed 4 × 8 array exhibits enhanced RCS as shown in Figure 12d.

The relationship between RCS enhancement range and phase compensation is similar to that of electromagnetic waves incident within the $Y=tan\;15^{{}^{\circ}}Z$ plane. When compensating the phase appropriately, the enhanced angle range is approximately ±$30^{{}^{\circ}}$, as depicted by the blue curve.

### 3.4. Manufacturing and Measurement

In order to confirm the effectiveness of the proposed RCS enhancement surface, we manufactured a physical sample and conducted monostatic RCS testing on the surface in a microwave chamber, which is depicted in the test scenario shown in Figure 13.

Before testing the electromagnetic surface, we need to obtain the phase of the phase shifter in the electromagnetic surface as a function of the excitation voltage. Therefore, we separately processed a phase shifter and measured its phase and insertion loss as a function of voltage, as shown in Figure 14.

We measured the phase of the phase shifter corresponding to multiple different voltages. Then, we used the least squares method to perform linear regression on the sampling points and obtained the relationship between the phase of the phase shifter and the applied excitation voltage, that is, $Phase(deg)=14.78BiasVoltage(V)-195.48^{{}^{\circ}}$.

The difference between the monostatic RCS of the electromagnetic surface and that of an equally sized metallic plate is equal to the difference in S21 values between the electromagnetic surface and the metallic plate. In other words, the magnitude of RCS enhancement on the surface is equivalent to the magnitude of enhancement observed in the S21 measurements. Therefore, here are the S21 results obtained from the testing.

Figure 15a shows the S21 measurement results of the sample under TE-polarized electromagnetic waves at the Phi = $0^{{}^{\circ}}$ plane. It is apparent that compared to the PEC plane, the designed surface exhibits superior RCS enhancement capability within the $Theta=\pm 45^{{}^{\circ}}$ range. The angle range for RCS enhancement tested is smaller than the simulated results, which is potentially due to the higher antenna gain and narrower beamwidth of the fabricated antenna, resulting in poorer radiation capability at larger angles. Displaying in Figure 15b are the S21 measurements of the sample under TM-polarized electromagnetic waves at the Phi = $90^{{}^{\circ}}$ plane. The surface has the capability to adjust the angle of RCS enhancement by manipulating the phase shifter. The figure depicts the S21 measurements taken as the surface enhances at various angles, such as Theta = $0^{{}^{\circ}}$, $15^{{}^{\circ}}$, $30^{{}^{\circ}}$, $45^{{}^{\circ}}$, and $60^{{}^{\circ}}$. Through demonstrating excellent RCS enhancement capability, the surface’s controllable feature of RCS enhancement direction is confirmed by the test results.

During the upcoming RCS testing process, the antenna will maintain vertical polarization while tilting the electromagnetic surface by 15 degrees. The turntable will rotate horizontally to measure the incidence of electromagnetic waves on the electromagnetic surface within the $Y=tan\;15^{{}^{\circ}}Z$ plane. The results of the monostatic RCS on the surface are shown in Figure 15c, indicating a very close resemblance between the measurement and simulation results. The red curve represents the surface’s RCS within the plane without compensation, while the blue curve represents the surface’s RCS with moderate compensation. It can be observed that moderate compensation can expand the enhanced angle range. The green curve represents the surface’s RCS with excessive compensation, where the RCS enhancement effect deteriorates near α = $0^{{}^{\circ}}$ due to phase mismatch. When the electromagnetic wave is incident on the electromagnetic surface within the $Y=tan\;30^{{}^{\circ}}Z$ plane, the test results for the monostatic RCS of the surface are shown in Figure 15d. The pattern of the enhanced angle range and phase compensation situation is similar to the case of electromagnetic wave incidence within the $Y=tan\;15^{{}^{\circ}}Z$ plane. The blue curve represents the surface’s RCS with moderate compensation, with an enhanced range of approximately ±$30^{{}^{\circ}}$.

Six designed electromagnetic surfaces are attached around the circumference of a plastic thin-walled cylindrical body, with the cylinder’s longitudinal axis tilted at a 15-degree angle, then the RCS around the cylinder is measured.

The testing setup is illustrated in Figure 16a, and the results are shown by the red curve in Figure 16b. When equally sized PEC metal plates are attached around the cylinder, the measured RCS is represented by the black curve in Figure 16b. The comparison of the two curves shows that a good RCS enhancement effect has been achieved on the designed surface, providing a 10 dB higher RCS enhancement effect than PEC. The test results indicated by the red curve appear significant fluctuations, which is attributed to the interference of electromagnetic waves reflected by adjacent electromagnetic surfaces, and this led to constructive or destructive interference at various angles.

## 4. Conclusions

This study introduces a novel controllable orientation radar cross section (RCS) enhancement surface. By designing and fabricating one-dimensional and two-dimensional Van Atta array structures and conducting tests, we have successfully confirmed the feasibility and effectiveness of this technology. Experimental results demonstrate that the installation of six controllable orientation Van Atta arrays on the outer surface of a cylindrical barrier leads the performance index to over 10 dB RCS enhancement compared to PEC. This suggests that our proposed approach can significantly enhance the radar system’s detection performance of roadway obstacles, providing strong support for the development and safety capabilities of autonomous driving systems. Future efforts could focus on further optimizing the design, expanding application scenarios to achieve more intelligent and reliable autonomous driving technology. In addition, the advantages and disadvantages of the designed electromagnetic surface compared to other types of RCS-enhanced surfaces are shown in Table 3.

## Figures and Tables

**Figure 1 sensors-25-04048-f001:**
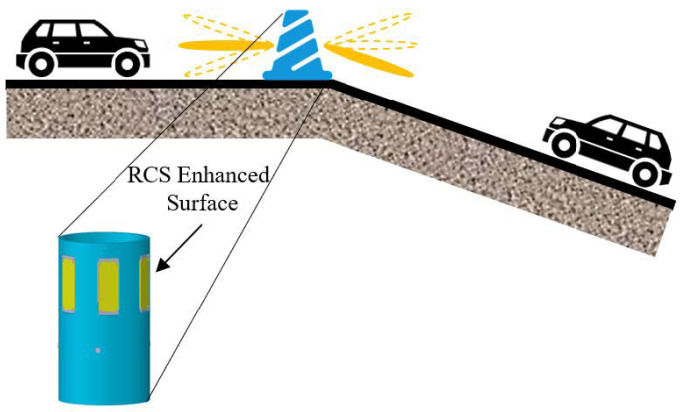
Application scenarios of RCS-enhanced surfaces in autonomous driving technology.

**Figure 2 sensors-25-04048-f002:**
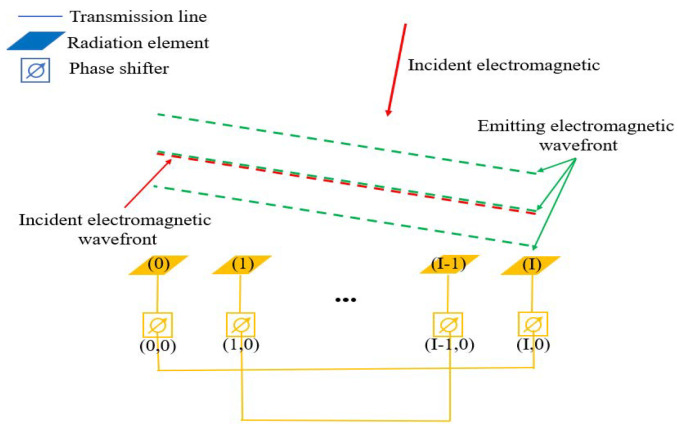
One-dimensional electromagnetic retroreflective array with adjustable wavefront phases.

**Figure 3 sensors-25-04048-f003:**
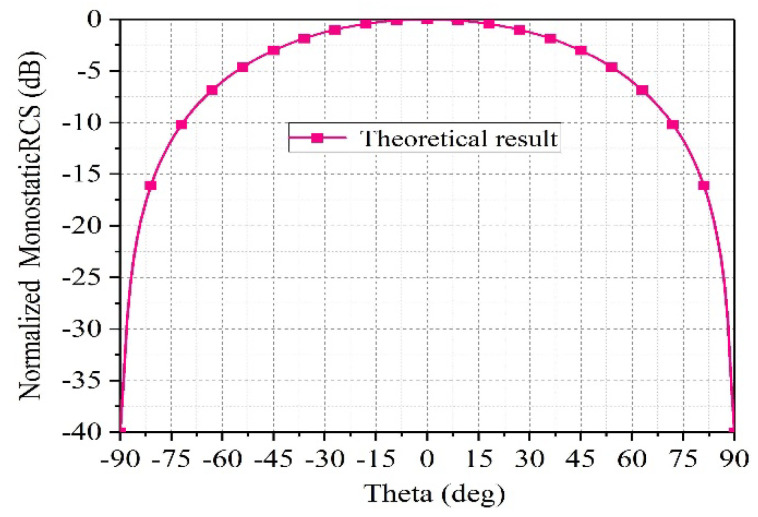
The theoretical results of normalized monostatic RCS for one-dimensional arrays.

**Figure 4 sensors-25-04048-f004:**
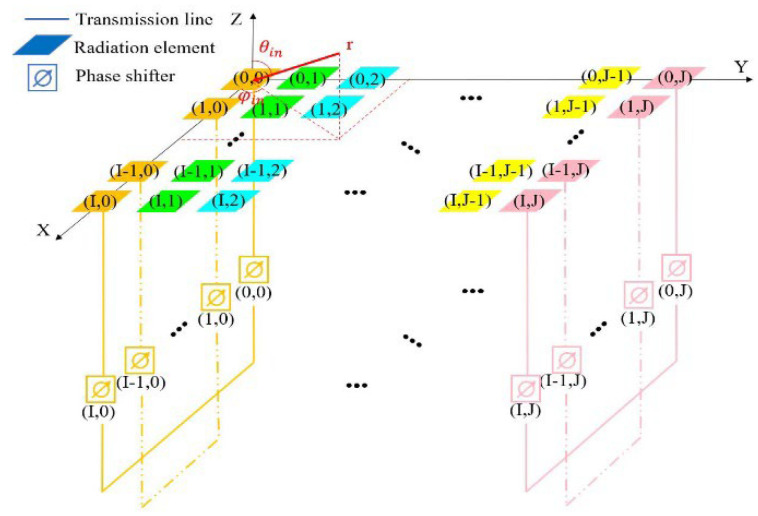
Schematic diagram of the RCS enhancement principle of the two-dimensional array.

**Figure 5 sensors-25-04048-f005:**
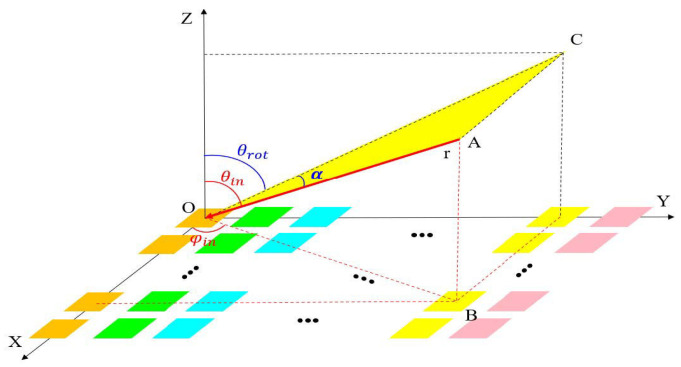
Schematic diagram of coordinate conversion.

**Figure 6 sensors-25-04048-f006:**
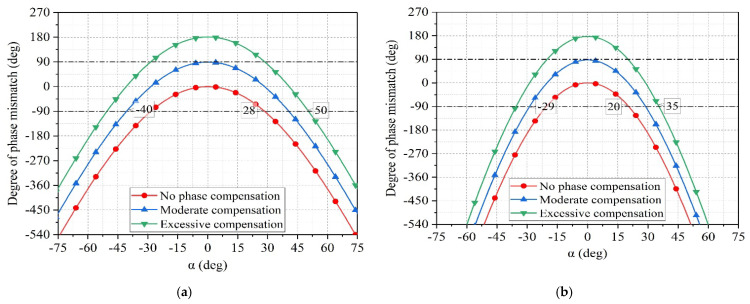
(**a**) The relationship between the degree of phase mismatch and the incident angle α when the incident wave is in the Y = tan 15°Z plane. (**b**) The relationship between the degree of phase mismatch and the incident angle α when the incident wave is in the Y = tan 30°Z plane.

**Figure 7 sensors-25-04048-f007:**
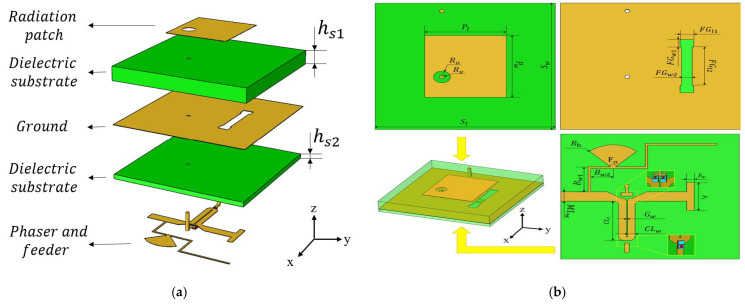
(**a**) Unit structure (exploded view). (**b**) Unit structure and key dimensions.

**Figure 8 sensors-25-04048-f008:**
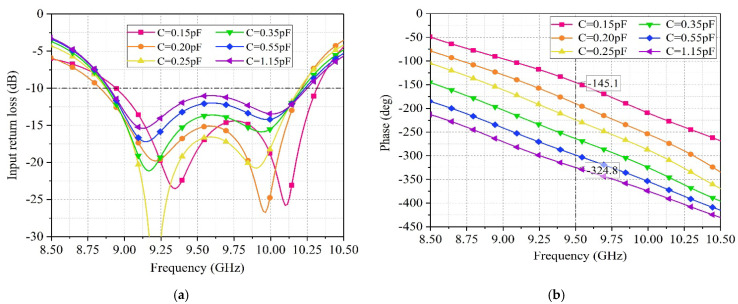
(**a**) Return loss of the antenna element at different capacitance values of varactor diodes. (**b**) The phase of electromagnetic waves emitted by the antenna unit at different capacitance values of the varactor diode.

**Figure 9 sensors-25-04048-f009:**
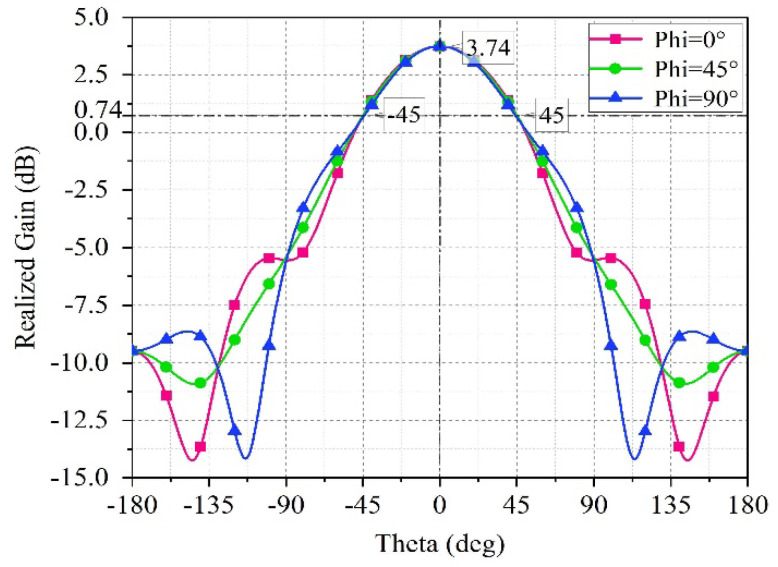
Antenna element pattern.

**Figure 10 sensors-25-04048-f010:**
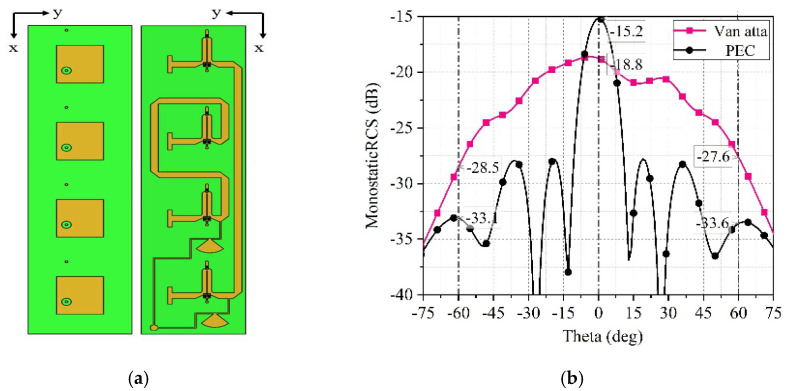
(**a**) 1 × 4 Van Atta array structure. (**b**) RCS enhancement in the Phaser = 0°plane.

**Figure 11 sensors-25-04048-f011:**
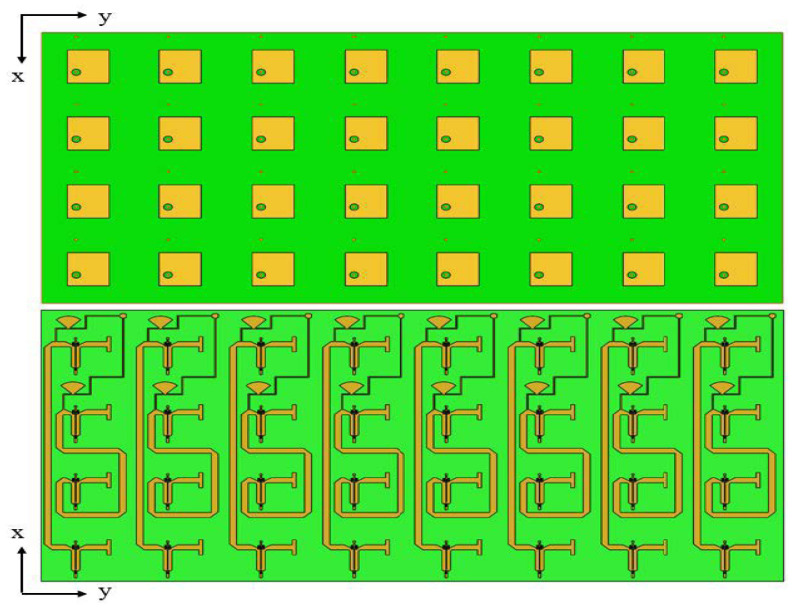
Two-dimensional 4 × 8 adjustable RCS enhancement surface.

**Figure 12 sensors-25-04048-f012:**
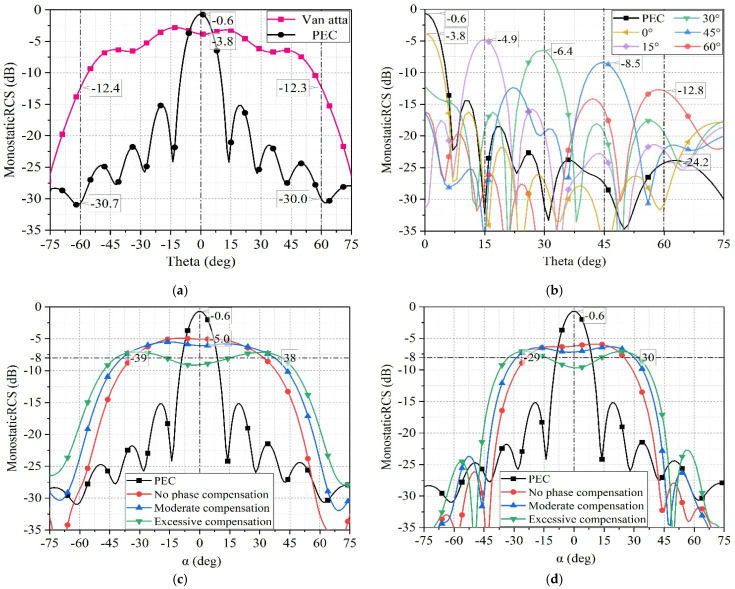
(**a**) The monostatic RCS of the proposed 4 × 8 array in the X-O-Z plane. (**b**) The monostatic RCS of the proposed 4 × 8 array in the Y-O-Z plane. (**c**) The monostatic RCS of the proposed 4 × 8 array in the Y = tan 15°Z plane. (**d**) The monostatic RCS of the proposed 4 × 8 array in the Y = tan 30°Z plane.

**Figure 13 sensors-25-04048-f013:**
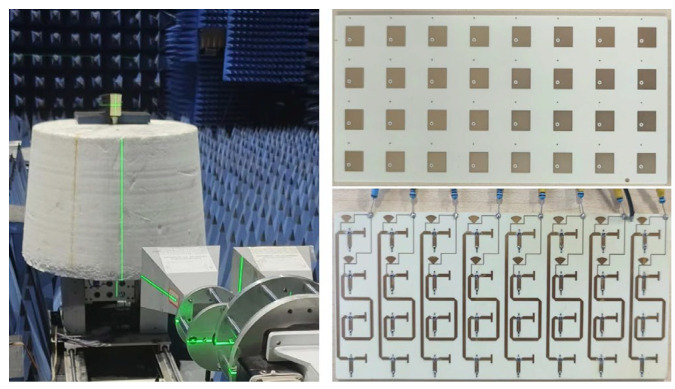
Physical fabrication and testing of the prototype.

**Figure 14 sensors-25-04048-f014:**
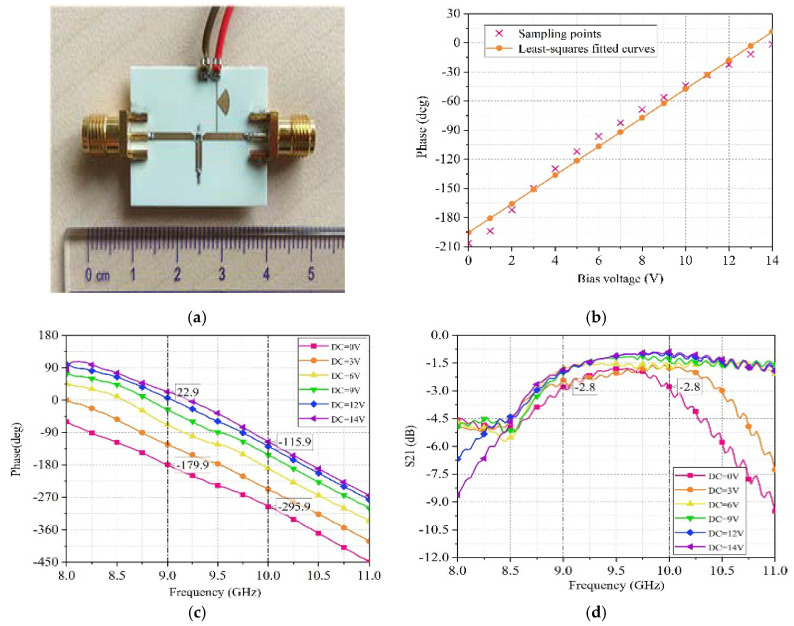
(**a**) Phase shifter. (**b**) The Relationship between phase shifter phase and bias voltage. (**c**) Measurement results of phase variation with bias voltage. (**d**) Measurement results of insertion loss varying with bias voltage.

**Figure 15 sensors-25-04048-f015:**
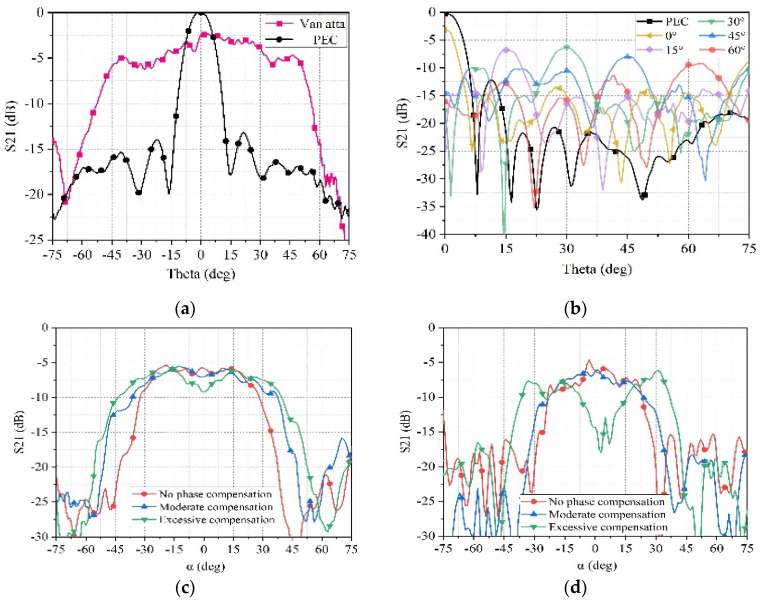
(**a**) The measured S21 of the 4 × 8 array in the X-O-Z plane. (**b**) The measured S21 of the 4 × 8 array in the Y-O-Z plane. (**c**) The measured S21 of the 4 × 8 array in the Y = tan 15°Z plane. (**d**) The measured S21 of the 4 × 8 array in the Y = tan 30°Z plane.

**Figure 16 sensors-25-04048-f016:**
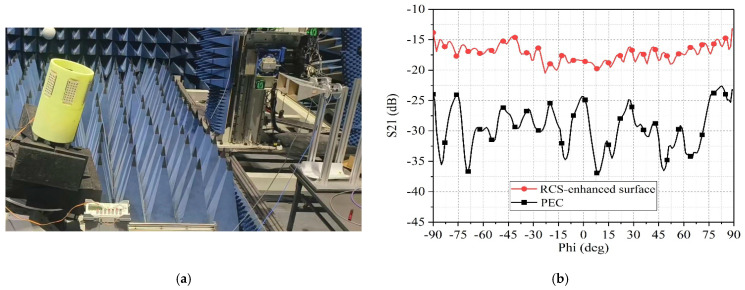
(**a**) RCS test scenario of cylindrical obstacle. (**b**) The measured S21 of the cylindrical obstacle.

**Table 1 sensors-25-04048-t001:** Key dimensions of the proposed antenna unit (Unit: mm).

Symbol	Value	Symbol	Value
$P_{l}$	6.77	$F_{w}$	0.62
$P_{w}$	7.33	${CL}_{l}$	4.7
$S_{l}$	15	${CL}_{w}$	0.64
$S_{w}$	15	$G_{w}$	0.1
$R_{o}$	0.7	${ML}_{w}$	1.12
$R_{v}$	0.2	$B_{w1}$	3.5
${FG}_{l1}$	1.14	$B_{w2}$	1.44
${FG}_{w1}$	0.88	$F_{a}$	$44^{{}^{\circ}}$
${FG}_{l2}$	4.55	$F_{b}$	2.84
${FG}_{w2}$	0.88	$h_{s1}$	1.524
$F_{l}$	3.35	$h_{s2}$	0.508

**Table 2 sensors-25-04048-t002:** The phase of each column’s phase shifter in the 4 × 8 array varies in different enhancement states when electromagnetic waves are incident within the Y = tan 15°Z plane.

States		No Phase Compensation	Moderate Compensation	Excessive Compensation
The phase gradient		$44.26^{{}^{\circ}}$	$38.93^{{}^{\circ}}$	$33.51^{{}^{\circ}}$
Phase of phase shifter (Unit: degree)	1	${-195.0}^{{}^{\circ}}$	${-232.3}^{{}^{\circ}}$	${-270.2}^{{}^{\circ}}$
	2	${-239.2}^{{}^{\circ}}$	${-271.2}^{{}^{\circ}}$	${-303.7}^{{}^{\circ}}$
	3	${-283.5}^{{}^{\circ}}$	${-310.2}^{{}^{\circ}}$	${-157.3}^{{}^{\circ}}$
	4	${-147.8}^{{}^{\circ}}$	${-169.1}^{{}^{\circ}}$	${-190.8}^{{}^{\circ}}$
	5	${-192.0}^{{}^{\circ}}$	${-208.0}^{{}^{\circ}}$	${-224.3}^{{}^{\circ}}$
	6	${-236.3}^{{}^{\circ}}$	${-246.9}^{{}^{\circ}}$	${-257.8}^{{}^{\circ}}$
	7	${-280.5}^{{}^{\circ}}$	${-285.9}^{{}^{\circ}}$	${-291.3}^{{}^{\circ}}$
	8	${-324.8}^{{}^{\circ}}$	${-324.8}^{{}^{\circ}}$	${-324.8}^{{}^{\circ}}$

**Table 3 sensors-25-04048-t003:** Comparison of different RCS enhancement schemes.

RCS Enhancement Scheme	Advantage	Disadvantage	RCS Enhancement Effect
Corner reflector [13]	Simple structure, wide bandwidth [14]	Large volume, single electromagnetic characteristics [15]	7~10 dB
Luneberg lens [16]	Good RCS enhancement effect [17]	3D structure, large volume [18]	7~10 dB
Van Atta array [10]	Wide bandwidth, wide angle, low profile [19]	Electromagnetic characteristics are uncontrollable [20]	10~20 dB
PON array [21]	Conformal.	Extremely narrow bandwidth [22]	10 dB
Active array with added amplifier [23]	Good RCS enhancement effect [24]	High energy consumption [25]	20~30 dB
Adjustable phase gradient surface [26]	Adjustable direction of RCS enhancement [27] Conformal.	Requires a large number of control channels and high costs [28]	20 dB
The proposed surface [29]	Adjustable direction of RCS enhancement wide bandwidth, low profile, [30] conformal low energy consumption [31]	Need to know the direction of the incoming wave.	10~15 dB

## Data Availability

The original contributions presented in this study are included in the article.

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
