# Peer review of "Orientation Controllable RCS Enhancement Electromagnetic Surface to Improve the Road Barriers Detectability for Autonomous Driving Radar"

_sensors, 2025, doi:10.3390/s25134048_

Round 1
Reviewer 1 Report
Comments and Suggestions for Authors
In the manuscript under review, the authors investigated theoretically using formal and Ansys HFSS modeling and experimentally the 6-orientation controllable Van Atta array, which can be used to improve the road pile detectability of on-board microwave radar for autonomous driving system that is the important direction of the development of the current automobile industry. This time, they arranged the one-dimensional Van Atta array in another dimension forming a two-dimensional array, and enabling adjustable orientation of radar cross section (RCS) enhancement due to the controllable phase of the reflected electromagnetic waves. In the result, the proposed electromagnetic backtracking surface has the following advantages: (1) The horizontal RCS enhancement angle range is very wide, and it only reflects high gain echoes in the direction of the incoming wave, without interference to other vehicles in the horizontal direction. (2) The vertical direction can adjust the angle to adapt to various road conditions, and the adjustment will not interfere with wireless devices at other heights. (3) The studied electromagnetic surface is relatively thin, making it easy to form cylindrical or conical shapes. To my mind, the manuscript is well written including a clear application scenarios of RCS-enhanced surfaces in autonomous driving technology, description of the essence, correctness and advantages of the proposed approach. As an advantage of the manuscript in terms of the quality of the presentation, I note the table provided in the Conclusion section, which compares the advantages and disadvantages of the proposed and known solutions.
Therefore, I recommend accepting this manuscript after minor revision related to the readability and typos.
1) Introduction section. As is standard for scientific articles, this section ends with a brief description of the contents of the following sections.
2) Lines 14, 17, 21, 420, 422. “Van-Atta”. According to literary sources, it is written without a hyphen, since it is the name of the inventor. This spelling is also present in the text of the manuscript.
3) Line 67. Typo: “horizontal” instead of “hori-zontal”
4) Line 92. Typo: “array” instead of “ar-ray”
5) Lines 290, 375, and so on. “PEC”. The abbreviation is not deciphered.
6) Lines 249, 286, 312, 328. “HFSS”. The abbreviation is not deciphered. I think it will be clearer that this is an abbreviated name for a well-known antenna simulator if you will write “Ansys HFSS”
7) In my opinion, authors should carefully check References section for the correctness of the cited sources and typos. For example, I could not find a source of Reference 1. What is this?
Author Response
Comments 1: Introduction section. As is standard for scientific articles, this section ends with a brief description of the contents of the following sections.
Response 1: The article is organized as follows. Section II discusses the principles of RCS Enhancement. In Section III, the proposed surface is fabricated and measured. Finally, Section IV concludes this article.
Comments 2: Lines 14, 17, 21, 420, 422. “Van-Atta”. According to literary sources, it is written without a hyphen, since it is the name of the inventor. This spelling is also present in the text of the manuscript.
Response 2: Thanks for your suggestion. We have made the necessary modifications as requested.
Comments 3: Line 67. Typo: “horizontal” instead of “hori-zontal”
Response 3: Thank you for your suggestion. We have made the necessary modifications as requested.
Comments 4: Line 92. Typo: “array” instead of “ar-ray”
Response 4: Thank you for your suggestion. We have made the necessary modifications as requested.
Comments 5: Lines 290, 375, and so on. “PEC”. The abbreviation is not deciphered.
Response 5: Thanks for your question, PEC is Perfect Electric Conductor.
Comments 6: Lines 249, 286, 312, 328. “HFSS”. The abbreviation is not deciphered. I think it will be clearer that this is an abbreviated name for a well-known antenna simulator if you will write “Ansys HFSS”
Response 6: Thank you for your suggestion. We have made the necessary modifications as requested.
Comments 7: In my opinion, authors should carefully check References section for the correctness of the cited sources and typos. For example, I could not find a source of Reference 1. What is this?
Response 7: Thank you very much for your suggestion, because the Chinese journal has been translated into English, which may be inaccurate and cause difficulties in finding. If you are interested in this article, you can search the topic on CNKI: "Status and trends of global autonomous vehicle applications".
Reviewer 2 Report
Comments and Suggestions for Authors
The manuscript introduces a technically sound and well-motivated approach for improving the detectability of roadside obstacles using a novel orientation-controllable radar cross-section (RCS) enhancement surface. By combining a modified one-dimensional Van Atta array with vertical phase control in a 2D layout, the authors enable backscattering in programmable directions. The concept addresses a critical challenge in autonomous driving radar: ensuring reliable detection of low-RCS targets such as plastic barriers and billboards across various road topographies.
The paper is structured clearly, and the progression from theoretical derivation to practical design and full experimental validation is effective. The use of a varactor-based analog phase shifter with column-wise bias control offers a low-complexity method to steer the RCS enhancement direction without significantly increasing system cost or complexity. The measured RCS results, both for planar and cylindrical implementations, support the authors’ claims of controllability and enhancement gain.
That said, the following points should be addressed to strengthen the manuscript and ensure it meets the highest standard for clarity, completeness, and contextual depth:
- While the paper explains that each column of the array uses identical biasing and achieves 0–360° coverage via dual-phase shifters, the quantization or resolution of achievable phase states is not discussed. Are there constraints introduced by diode tuning nonlinearity, thermal drift, or bias line loading? Please clarify how precisely the target phase profiles can be synthesized and maintained.
- The use of a single bias control per column is elegant and minimizes wiring complexity, but practical deployment would require a scalable control interface. A short discussion on how this can be integrated into real vehicle platforms (e.g., microcontroller, wireless bias control, or self-sensing feedback) would improve the application scope.
- Some inconsistencies between simulation and measurement results (notably the narrower enhancement angles and fluctuations in Fig. 15) are acknowledged, but further elaboration would help. Can the authors discuss in more depth how element-to-element variation, feeding structure imperfections, or mutual coupling may have contributed?
- The manuscript cites foundational and recent works on Van Atta arrays and metasurfaces, but it would benefit from inclusion of more recent literature on programmable and beam-steering RCS enhancement surfaces, particularly those applying compact reconfigurable array concepts. In this regard, the following references are technically relevant and can be cited in the introduction to enrich the discussion on array-based beam shaping, retrodirective response control, and near-field design:
-
Battaglia et al., 2023: “Four-Beams-Reconfigurable Circular-Ring Array Antennas for Monopulse Radar Applications,” Radio Science, DOI: 10.1029/2023RS007776. This design shares themes of reconfigurable beam control in circular array geometries, which aligns closely with the authors’ cylindrical surface implementation.
This is a well-executed and practically relevant contribution that offers measurable improvement to real-world radar visibility of passive roadside infrastructure. With minor revisions—especially expanded contextual discussion and language refinement—the manuscript will be a valuable addition to the literature on smart electromagnetic surfaces and autonomous sensing systems.
Author Response
Comments 1: While the paper explains that each column of the array uses identical biasing and achieves 0–360° coverage via dual-phase shifters, the quantization or resolution of achievable phase states is not discussed. Are there constraints introduced by diode tuning nonlinearity, thermal drift, or bias line loading? Please clarify how precisely the target phase profiles can be synthesized and maintained.
Response 1: This is a good question. This article uses a phase shifter based on a varactor diode, and the phase of the phase shifter is determined by the equivalent capacitance of the varactor diode, which is related to its bias voltage. Therefore, different bias voltages can be applied to the phase shifter to obtain different transmission phases. The design of this article is implemented using the DAC0832LCN chip in conjunction with the operational amplifier LM358 (DAC0832 is a current type DAC chip that needs to be converted into voltage through an operational amplifier), as shown in Figure R1. DAC0832 is an 8-bit DAC chip, which can provide 2 ^ 8=256 different voltage levels, so the phase resolution is 360/256 ≈ 1.4 °.
Due to the reverse bias of the varactor diode, the current is very small, so the heat generation is also very small. To be honest, we did not suffer from heat generation issues during the design and testing process.
Due to the nonlinearity of diode tuning, there is a certain phase error. In order to control the phase as accurately as possible, we had to separately process the phase shifter and measure the transmission phase at different voltages, obtaining the sampling points shown in Figure R2. I believe that the relationship between voltage and phase is approximately linear, so I used the least squares method to linearly fit the phase=14.78 × excitation voltage (V) -195.48 °. In fact, I also tried to fit the phase and excitation voltage relationship of the second-order function according to the second-order function, but through the actual measurement of the array, we found that the test results obtained by linear fitting are closer to the simulation results, so we finally adopted linear fitting. I think if someone's design has a phase to voltage relationship obtained through sampling that is closer to a second-order function, then fitting the phase to voltage relationship using a second-order function may be more suitable for their design.
Finally, due to the simultaneous transmission of DC energy and microwave energy on the microstrip line, in order to prevent microwave energy from leaking through the DC bias line and affecting transmission loss and phase, a fan-shaped branch was added to the DC bias line, which works well and effectively isolates microwave and DC energy. When designing a fan-shaped branch, the distance between the fan-shaped branch and the microstrip line is about a quarter wavelength, the fan-shaped radius is slightly less than a quarter wavelength, and the fan-shaped angle is around 90 °.
R.fig1 Typical Application(DAC0832LCN)
R2
R3
Comments 2: The use of a single bias control per column is elegant and minimizes wiring complexity, but practical deployment would require a scalable control interface. A short discussion on how this can be integrated into real vehicle platforms (e.g., microcontroller, wireless bias control, or self-sensing feedback) would improve the application scope.
Response 2:Using a single bias control for each column is elegant and minimizes wiring complexity, but practical deployment requires scalable control interfaces. A brief discussion on how to integrate it into real vehicle platforms such as microcontrollers, wireless bias control, or self sensing feedback will expand its application scope.
Thank you for your question. In the design, in order to enhance scalability, we use serial data transmission between modules, and ultimately consolidate the power and data lines into one MCU controller, as shown in the figure below. Since the control program has already been written in the MCU, all that is needed to control all the electromagnetic surfaces connected in series on the control board is to send an RCS enhancement direction angle to the MCU through the serial port. If an electromagnetic surface needs to be added, it only needs to continue to be connected in series, which has high scalability (but it should be considered that too many serial stages will cause the transmission line to be too long, resulting in poor signal quality and insufficient driving ability). In addition, by adding other sensors such as WiFi and wireless transmission modules to the MCU, it is possible to wirelessly and remotely control the electromagnetic surface.
R4 DAC control board and MCU
Comments 3: Some inconsistencies between simulation and measurement results (notably the narrower enhancement angles and fluctuations in Fig. 15) are acknowledged, but further elaboration would help. Can the authors discuss in more depth how element-to-element variation, feeding structure imperfections, or mutual coupling may have contributed?
Response 3: Thank you for your question, it's a very good one. Firstly, from the testing site shown in Figure 13a, it can be observed that our testing system is actually a quasi single station RCS. Since one horn cannot simultaneously transmit and receive, we placed two horns very close to simulate the situation of a single station RCS. Obviously, the electromagnetic wave emitted by one horn is reflected on the array and received by the adjacent horn. This results in an angle between the measured reflected wave and the actual reflected wave. We have kept the distance between the horn and the test piece far enough to minimize the impact of this angle, However, there is still an inevitable measurement error (note that this testing method can only be used when the array is small and the beam width is wide. If the array is very large and the beam width is narrow, the small distance between the receiving and transmitting horns may also cause the electromagnetic waves reflected by the array to be unable to be measured by the receiving horn. For example, if the array gain is very high and the beam width is only 1 °, the electromagnetic waves reflected by the array will return to the transmitting antenna along the incident direction. Due to the narrow beam, the adjacent receiving horn cannot measure the reflected waves. In addition, the two horns cannot be too close, otherwise the energy of the transmitting horn will be directly coupled to the receiving horn, resulting in incorrect test results). In addition to measurement system errors, due to errors between antennas, There is a certain coupling due to the close distance.
Another difficult issue is that the array cannot be calibrated at 0 degrees. Due to the fact that both ends of the phase shifter are connected to antennas, it is impossible to measure the transmission phase of the phase shifter, and even small deviations in the soldering position of the varactor diode may cause differences in the transmission phase of the phase shifter. In fact, during the design process, we used manual soldering of diodes in the first version. Due to the large manual soldering error, the initial phase difference of the phase shifter was also significant, and the electromagnetic surface working state was poor. In the second version, we used SMT surface mount to solder the varactor diode, minimizing the transmission phase difference between phase shifters by ensuring consistent soldering. I have discussed with many designers who work on adjustable electromagnetic surfaces how to solve the problem of calibrating the 0-position of adjustable electromagnetic surfaces, but I have not yet received a satisfactory solution.
Comments 4:The manuscript cites foundational and recent works on Van Atta arrays and metasurfaces, but it would benefit from inclusion of more recent literature on programmable and beam-steering RCS enhancement surfaces, particularly those applying compact reconfigurable array concepts. In this regard, the following references are technically relevant and can be cited in the introduction to enrich the discussion on array-based beam shaping, retrodirective response control, and near-field design:
- Battaglia et al., 2023: “Four-Beams-Reconfigurable Circular-Ring Array Antennas for Monopulse Radar Applications,” Radio Science, DOI: 10.1029/2023RS007776. This design shares themes of reconfigurable beam control in circular array geometries, which aligns closely with the authors’ cylindrical surface implementation.
Response 4: Thank you very much for your suggestion. I have read this article and will also cite it in my submitted paper.
Reviewer 3 Report
Comments and Suggestions for Authors
The authors propose a novel approach for radar cross section enhancement surface design and prove its effectiveness by fabrication and measurements. They use one- and two- dimensional controllable orientation Van Atta array structures on the surface of a cylindrical barrier in order to improve radar detection of roadway obstacles, which is very important in safety of autonomous driving systems. In comparison with other relevant works, this structure achieves many advantages.
The paper is very well written and the contribution is very clear. There are just minor text editing changes to be revised throughout the paper, for example 'require-mints'; it is usual to use subscripts for parameters like for the phase of phase shifter instead of 'phiPhaser', and big letters for 'Phaser' in sentences; please add missing word at the end of sentence: ' ... to over 10dB RCS enhancement compared to PEC. ', like '... compared to PEC surface'. Also, if possible, please comment on bistatic RCS detection in these applications.
Author Response
Comments 1: There are just minor text editing changes to be revised throughout the paper, for example 'require-mints'; it is usual to use subscripts for parameters like for the phase of phase shifter instead of 'phiPhaser', and big letters for 'Phaser' in sentences; please add missing word at the end of sentence: ' ... to over 10dB RCS enhancement compared to PEC. ', like '... compared to PEC surface'. Also, if possible, please comment on bistatic RCS detection in these applications.
Response 1: Thanks for your suggestion. We have made the necessary modifications as requested.